# Cooperative dynamic polaronic picture of diamond color centers

Takuto Ichikawa [1,4], Junjie Guo [1], Paul Fons [2], Dwi Prananto [3], Toshu An [3] & Muneaki Hase [1] ✉

Polarons can control carrier mobility and can also be used in the design of quantum devices. Although much effort has been directed into investigating the nature of polarons, observation of defect-related polarons is challenging due to electron-defect scattering. Here we explore the polaronic behavior of nitrogen-vacancy (NV) centers in a diamond crystal using an ultrafast pump-probe technique. A 10-fs optical pulse acts as a source of high electric field exceeding the dielectric breakdown threshold, in turn exerting a force on the NV charge distribution and polar optical phonons. The electronic and phononic responses are enhanced by an order of magnitude for a low density of NV centers, which we attribute to a combination of cooperative polaronic effects and scattering by defects. First-principles calculations support the presence of dipolar Fröhlich interaction via non-zero Born effective charges. Our findings provide insights into the physics of color centers in diamonds.

The concept of a polaron is that of a free carrier accompanied by a "phonon cloud", a quasiparticle which consists of an electron and a phonon in solids[1]. The formation of a polaron is characterized by the electron-phonon coupling; the size of the Fröhlich polaron is more than several lattice constants[2], while that of the Holstein polaron is on the order of the lattice constant[3]. Since the polarons propagate at a lower velocity than free carriers in materials, the existence of the Fröhlich polaron can be characterized by a larger carrier mass, $m_{e,h} \cong m_{e,h}^*\left(1+\frac{\alpha}{6}\right)$, where $\alpha$ is the Fröhlich electron-phonon coupling constant and $m_{e,h}^*$ is the free carrier ($e$: electron or $h$: hole) mass[1,2].

On the contrary, a different type of polaron, the Jahn-Teller (JT) polaron is produced around degenerate orbitals, which induces the local structure to deform to attain a lower potential energy[4,5]. Recent studies have investigated JT effects on diamond color centers[6], in particular the nitrogen-vacancy (NV) center[7], which has attracted considerable attention owing to its potential applications in quantum sensing[8,9], biomedicine[10], and quantum information sciences[11,12]. Within the band gap of the diamond NV, the $^3A_2$ electronic ground and doubly degenerate $^3E$ excited states

are optically coupled by a zero-phonon line (ZPL) transition at 1.95 eV (ref. [13]). As a result of its orbital degeneracy, the $^3E$ state couples to a doubly degenerate vibrational mode of $e$-symmetry to form an $E \otimes e$ JT system. The JT-active mode involves the displacement of the carbon atoms that surround the vacancy. Although the $E \otimes e$ JT system has been examined previously[13,14], there remains the possibility of other types of polarons emerging due to symmetry breaking in the NV local structure (Fig. 1a inset)[15].

In this work, we examine the polaronic picture of diamond NV centers using an ultrafast pump-probe technique with a 10-fs near-infrared optical pulse. We demonstrate that the intensity of electronic and phononic polarization responses show a dramatic increase with a 13-fold magnification for a dose level of $1.0 \times 10^{12}\,\text{N}^+\,\text{cm}^{-2}$ and with the electric field of the pump pulse close to the dielectric breakdown threshold $(1-2 \times 10^7\,\text{V cm}^{-1})$[16]. We attribute this large response to the combination of the cooperative polaronic effect[17] and scattering by defects[18]. First-principles calculations reveal the presence of nonlinear polarization around the NV center via non-zero Born effective charges supporting the Fröhlich nature of the polarons.

[1]Department of Applied Physics, Faculty of Pure and Applied Sciences, University of Tsukuba, Tsukuba, Ibaraki, Japan. [2]Department of Electronics and Electrical Engineering, Faculty of Science and Technology, Keio University, Yokohama, Kanagawa, Japan. [3]School of Materials Science, Japan Advanced Institute of Science and Technology, Ishikawa, Japan. [4]Present address: Sensing System Research Center, National Institute of Advanced Industrial Science and Technology, Tosu, Saga, Japan. ✉e-mail: mhase@bk.tsukuba.ac.jp

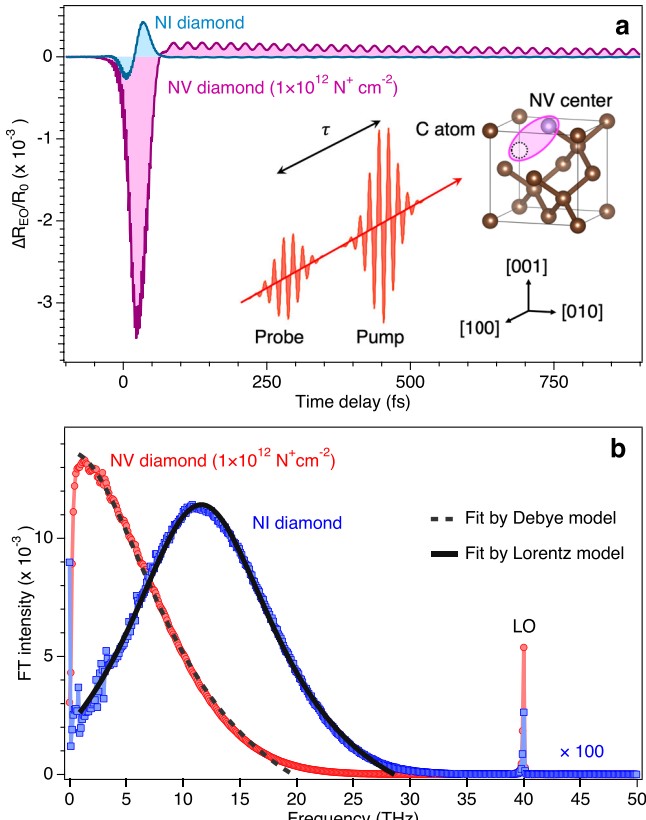

**Fig. 1 | Electro-optic response of NI and NV diamonds. a** Time-domain $\Delta R_{EO}/R_0$ signals for NI and NV diamonds obtained at the pump fluence of 1.3 mJ cm$^{-2}$. The time delay zero ($\tau$=0) was determined by the position of coherent artifacts (Supplementary Fig. 2). The inset represents the local structure of NV center (purple ellipse), together with the pump-probe method. **b** The Fourier transformed spectra obtained from the time-domain response in (**a**). The black dashed and solid lines are the fit by Debye and Lorentz models, respectively.

## Results

### Electro-optic response of NI and NV diamonds

Coherent phonons (CPs) in NV diamond were measured by a pump-probe electro-optic (EO) sampling technique at room temperature[19,20]. The NV diamond samples were prepared by $^{14}$N$^+$ ion implantation into four highly transparent electronic grade (EG) diamonds (the implantation doses were $2.0 \times 10^{11}$, $1.0 \times 10^{12}$, $5.0 \times 10^{12}$, and $2.0 \times 10^{13}$ N$^+$ cm$^{-2}$, respectively) grown by chemical vapour deposition (CVD) and subsequent annealing (see *Methods*)[21]. The light source was a near-infrared femtosecond oscillator with a central wavelength of $\backsim$800 nm (1.55 eV), a pulse duration of $\leq 10$ fs and a repetition rate of 75 MHz. Although the laser spectra extended from 660 nm (1.88 eV) to 940 nm (1.32 eV), the NV triplet-triplet transition ($^3A_2 \rightarrow ^3E$: 1.95 eV) does not generally occur[13]; thus, the $E \otimes e$ JT effect is expected to play a minor role, and coherent longitudinal optical (LO) phonons (0.165 eV) are excited without significant carrier excitation. On the other hand, there is the possibility of excitation of NV states via Urbach tails[22] associated with the presence of other defect states, such as P1 centers[23], which induce Urbach tails for a NV level with >1 eV broadening as can be seen in the density of states calculated by density functional theory (DFT) (Supplementary Fig. 1), enabling the absorption of laser light even when the excitation spectra extends up to 660 nm. In the present study, more importantly, the estimated electric field of the pump light was $\approx 1.4 \times 10^7$ V cm$^{-1}$ at 440 mW (or 1.3 mJ cm$^{-2}$), which is comparable to the dielectric breakdown threshold of diamond[16]. This means that under femtosecond laser irradiation, the NV centers are sufficiently

ionized to produce free electrons, even if optical transitions do not occur. Photoluminescence measurements indicate the electronic state of the NV diamond is a mixture of negatively charged states (NV$^-$) and the neutrally charged states (NV$^0$) because of the observation of the ZPL for NV$^-$ at 638 nm and broad peaks at $\backsim$680 nm, while the ZPL for NV$^0$ at 575 nm and a broad peak at $\backsim$660 nm, respectively[15,24].

The EO signals $\Delta R_{EO}/R_0$ for the non-implanted (NI) and NV diamonds ($1.0 \times 10^{12}$ N$^+$ cm$^{-2}$) observed at the pump fluence of 1.3 mJ cm$^{-2}$ are shown as a function of the time delay $\tau$ in Fig. 1a. Coherent artifacts are frequently observed at $\tau \approx 0$ fs in pump-probe experiments, but they are useful for determining the time delay zero[25,26]. We observed coherent artifacts in our EO signal up to $\backsim$100 fs (see Supplementary Fig. 2), showing $\tau \approx 0$ fs is slightly before the peak of the EO signal. The NI diamond exhibits a bipolar EO response, while the NV diamond shows a monopolar EO response on time scales <50 fs. The EO response of the NV diamond was enhanced by a factor of 13 over that of the NI diamond sample. Coherent oscillations were also observed after the transient EO response. To investigate these responses, Fourier transforms (FTs) of the EO signals were carried out as can be seen in Fig. 1b. Broad-band EO responses observed at frequencies lower than 25 THz could be characterized by Debye and Lorentz functions, suggesting hopping-like and Raman-like polarization are produced in the NV and NI diamonds, respectively[27]. The difference between the response frequency, i.e., a peak at $\backsim$12 THz for the NI and $\backsim$1 THz for the NV diamond, indicates faster electronic Raman polarization and slower hopping electronic polarization (or charge transfer), respectively, was generated by the pump pulse. In addition, the optical CP peak ($\approx$40 THz) appears for both NI and NV diamonds[28], while the FT intensity is dramatically amplified in the NV diamond case especially for the dose of $1.0 \times 10^{12}$ N$^+$ cm$^{-2}$ as described below.

### Coherent phonons of NI and NV diamonds

The coherent oscillation parts extracted from the EO sampling data are plotted for various $^{14}$N$^+$ doses in Fig. 2a. Each signal could be well fit by a damped harmonic oscillation function[29], $f(t) = A e^{-t/\tau_{LO}} \sin(\Omega_{LO} t + \psi)$, where $A$ is the initial amplitude, $\tau_{LO}$ and $\Omega_{LO}$ are the dephasing time and the frequency of the LO phonon, respectively, and $\psi$ is the initial phase. The four coefficients obtained by the fit show a dose dependence as shown in Fig. 2b–e. The frequency $\Omega_{LO}$ and dephasing time $\tau_{LO}$ observed for the NI diamond agree well with previous studies[28], and only very slight changes are observed in $\tau_{LO}$ and $\Omega_{LO}$ within experimental errors, which are mainly related to scattering factors involving defects[18]. On the other hand, the initial phonon amplitude, $A$, was found to be slightly larger for the lower dose of $2 \times 10^{11}$ N$^+$ cm$^{-2}$ and then was found to increase by about a factor of 13 for the NV diamond at $1.0 \times 10^{12}$ N$^+$ cm$^{-2}$ over the corresponding spectra of NI diamond; a monotonic decrease was found for higher dose levels. Thus, the $1.0 \times 10^{12}$ N$^+$ cm$^{-2}$ sample exhibited the maximum amplification (Fig. 2b). While the $\psi \backsim 0$ (sine-like) driving force dominates in the NI diamond sample due to impulsive stimulated Raman scattering (ISRS)[29], $\psi \backsim \pi/2$ is indeed observed for the NV diamond sample with a dose of $1.0 \times 10^{12}$ N$^+$ cm$^{-2}$, and was found to revert to $\psi \backsim 0$ again for the higher N$^+$ dose levels (Fig. 2e). From these results, a different cosine-like driving force other than ISRS is expected to be present in NV diamond. As mentioned above, the application of a strong electric field is equivalent to normal electronic excitation by visible light and coherent phonons are induced. Since electronic excitation is taking place, the initial phase has a cosine nature. Note that a cosine-like driving force was also observed in a Si crystal under near-resonance conditions, with an $E_0$ resonance of 3.4 eV (=365 nm), and a laser energy of 2.91–3.26 eV (380–425 nm)[30]. The possible photoexcitation of carriers using near-resonance light has been known to occur via Urbach tails in doped semiconductors[22]; this is not the case here as explained in the next section.

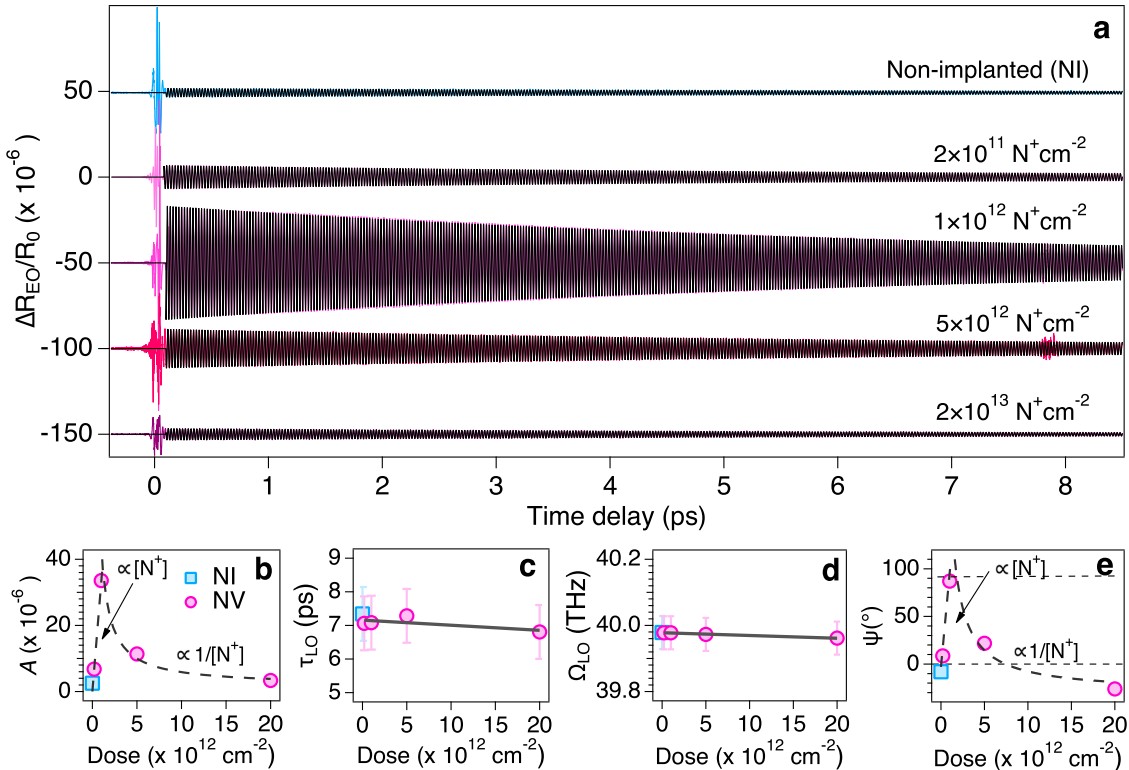

**Fig. 2 | Coherent phonon modulation of diamond by NV centers. a** Time-domain $\Delta R_{EO}/R_0$ signals for NI and NV diamonds obtained at the pump fluence of 1.3 mJ cm$^{-2}$ for various doses, [N$^+$]. The black line is the fit by a damped harmonic oscillator function from $\tau \approx 100$ fs where coherent artifacts disappear, and only coherent phonons (~40 THz) are observed. **b** Dose dependence of the initial amplitude $A$. The square blue marker indicates parameter for the NI diamond and the round pink markers for NV diamonds. The dashed line represents the fit by $A$ $\propto$ [N$^+$] for <1.0 × 10$^{12}$ N$^+$ cm$^{-2}$, while by $A \propto 1/$[N$^+$] for ≥1.0 × 10$^{12}$ N$^+$ cm$^{-2}$. **c** Dose dependence of the decay time $\tau_{LO}$. Error bars represent the standard deviations. **d** Dose dependence of the frequency $\Omega_{LO}$. Error bars represent the frequency resolution of ~0.1 THz when the range of time delay was ~9 ps. **e** Dose dependence of the initial phase $\psi$ of the coherent LO phonon extracted from the fitting in (**a**). The dashed line represents the fit by $\psi \propto$ [N$^+$] for <1.0 × 10$^{12}$ N$^+$ cm$^{-2}$, while by $\psi \propto 1/$[N$^+$] for ≥1.0 × 10$^{12}$ N$^+$ cm$^{-2}$.

## Generation mechanisms of coherent phonons in NI and NV diamonds

To address the generation mechanisms, the motion of the atomic displacement **Q** associated with the CP (LO mode) under irradiation by a femtosecond laser pulse can be described by ref. 31,

$$\mu\left(\ddot{\mathbf{Q}} + \frac{2}{\tau_{LO}}\dot{\mathbf{Q}} + \Omega_{LO}^2\mathbf{Q}\right) = \hat{\mathbf{e}}_i\left(\bar{R}\mathbf{E}_1\mathbf{E}_2 - \frac{4\pi Z^*}{\epsilon_\infty}\mathbf{P}^{NL}\right) \quad (1)$$

where $\mu$ is the effective mass of the atom, and $\hat{\mathbf{e}}_i$ is the unit displacement vector of the LO phonon, $\bar{R}$ is the Raman tensor, $\mathbf{E}_1(\mathbf{E}_2)$ is the electric field at the angular frequency $\omega_1$ ($\omega_2$) satisfying $\Omega_{LO} = \omega_2 - \omega_1$, $Z^*$ is the Born effective charge tensor[32], $\epsilon_\infty$ is the high-frequency dielectric constant, and

$$\mathbf{P}^{NL} = \epsilon_0\chi^{(2)}\mathbf{E}_1\mathbf{E}_2 + \int_{-\infty}^t J(t')dt' \quad (2)$$

is the macroscopic nonlinear polarization, where $\epsilon_0$ is the dielectric constant in vacuum, $\chi^{(2)}$ is the second-order nonlinear susceptibility tensor due to the NV centers, and $J(t)$ is the transient photocurrent, which is driven by the photoionization[33]. According to Eq. (1), the additional driving force $\mathbf{F}_{NV}$ in the NV diamond ($\chi^{(2)} \neq 0$) consists of the second term $\mathbf{F}_{NV} = -4\pi Z^*\mathbf{P}^{NL}/\epsilon_\infty$. In addition, this excitation scheme is often referred to as field-induced ISRS and is proportional to Re[$\chi^{(2)}$] due to the underlying nonlinear polarization[34], being consistent with the observed frequency dependence of the EO response of the Debye relaxation model[35] ($\propto \frac{1}{1+i\omega\tau_D}$, where $\omega$ is the angular frequency and $\tau_D$ is the relaxation time), shown in Fig. 1b.

According to this scenario, dipolar electron-phonon interaction, e.g., Fröhlich electron-phonon coupling[2], as characterized by the Born effective charge tensor $Z^*$ is expected to occur for the NV center, as revealed by nonlinear emission experiments[15]. Therefore, we investigated the characteristics of $Z^*$ with the framework of group theory and DFT simulations. The independent NV centers exhibit uniaxial anisotropy belonging to the crystallographic point group $3m(C_{3v})$ and the four possible orientations of the NV axis (Fig. 1a inset)[13]. According to these symmetry considerations, it can be concluded that force exists only along the [001] direction (i.e., $\mathbf{F}_{NV} = F_{NV}\mathbf{z}$) as given by the average product of $Z^*$ and $\chi^{(2)}$ for the four NV axis directions. The nonlocal photocurrent $J(t)$ is driven by photoionization with the pump electric field, i.e., followed by the first term of Eq. (2), and thus being along the same [001] direction. The magnitude of the first term of the force $\mathbf{F}_{NV}$ can then be expressed as,

$$F'_{NV} = -\frac{4\pi\epsilon_0}{\epsilon_\infty}\left(2E_xE_y\right)$$
$$\times \frac{\left(Z_{32}^* - \sqrt{2}Z_{22}^*\right)\left(\sqrt{2}\chi_{15}^{(2)} + 2\chi_{22}^{(2)}\right) + \left(\sqrt{2}Z_{23}^* - Z_{33}^*\right)\left(\chi_{31}^{(2)} - \chi_{33}^{(2)}\right)}{3\sqrt{3}}$$
$$(3)$$

To visualize the force $F'_{NV}$ given by Eq. (3), we have determined the values of $Z^*$ for the individual atoms in a diamond supercell containing a NV center using DFT. Figure 3a shows the calculated charge density distribution for a negatively charged NV center (NV$^-$). The charge density modulation extends from the NV$^-$ axis to about 0.4$a$ where $a$ is the lattice constant; C atoms with large absolute values of $Z^*$ are present. In the projection shown in Fig. 3b, the adjacent atoms are zigzag-

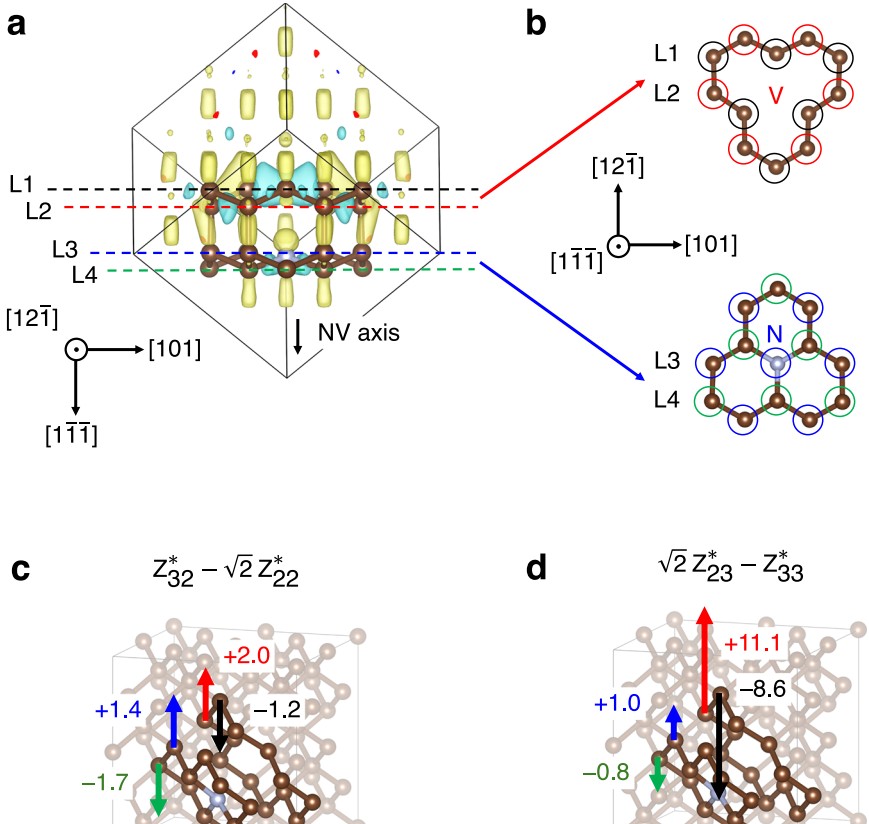

**Fig. 3 | Calculation of the born effective charge tensor in NV diamond. a** Charge density distribution of NV diamond (C atoms only partially shown) in side-view with respect to the NV axis $[1\bar{1}\bar{1}]$. The brown and silver spheres indicate C and N atoms, respectively. The positive (negative) charges inside and in the cross section of the supercell are expressed in light green (yellow) and dark blue (red), respectively. The dashed lines L1 (black), L2 (red), L3 (blue), and L4 (green) represent planes containing 6 or 7 atoms, respectively. **b** Depictions of NV diamond atoms in the two planes {L1, L2} or {L3, L4} in top view with respect to the NV axis $[1\bar{1}\bar{1}]$. The plane containing the atoms is represented by the colored circles corresponding to the dashed lines in (**a**). **c, d** The representations of the values $Z_{32}^* - \sqrt{2}Z_{22}^*$ and $\sqrt{2}Z_{23}^* - Z_{33}^*$, calculated using the sum of $Z^*$ of the atoms; $Z_L^*$ in plane L (=L1, L2, L3, L4). The semi-transparent background shows the supercells in basis {x∥[100], y∥[010], z∥[001]}. The arrows represent the sign and magnitude scale of the values, $Z_{32}^* - \sqrt{2}Z_{22}^*$ or $\sqrt{2}Z_{23}^* - Z_{33}^*$, and their colors correspond to the planes, i.e., L1 (black), L2 (red), L3 (blue), and L4 (green), in which the atoms are contained.

bonded, which results in the inclusion of 6 or 7 atoms in the paired planes {L1, L2} or {L3, L4} (Fig. 3b). The LO phonons are produced by atoms on the paired planes moving in the opposite $z$ directions. The sum of the $Z^*$ for the atoms, $Z_L^*$ in each plane L ( = L1, L2, L3, L4) for the basis {x ∥ [100], y ∥ [010], z ∥[001]} are:

$$Z_{L1}^* = \begin{pmatrix} 0.09526 & 2.36922 & -1.2385 \\ 2.94794 & -1.70272 & -3.36511 \\ -0.90689 & -3.6066 & 3.87162 \end{pmatrix} \quad (4)$$

$$Z_{L2}^* = \begin{pmatrix} 0.49421 & -2.72356 & 2.06925 \\ -3.24958 & 1.85964 & 4.41104 \\ 1.76788 & 4.63052 & -4.81366 \end{pmatrix} \quad (5)$$

$$Z_{L3}^* = \begin{pmatrix} -0.88598 & -0.01077 & 0.08163 \\ -0.51741 & -0.91526 & -0.11213 \\ -0.2087 & 0.0993 & -1.12657 \end{pmatrix} \quad (6)$$

$$Z_{L4}^* = \begin{pmatrix} 1.59242 & -0.04782 & 0.25591 \\ -0.14129 & 1.40252 & 0.22134 \\ 0.2024 & 0.26036 & 1.14698 \end{pmatrix} \quad (7)$$

Using tensor components presented in Eqs. (4)–(7), the values of $Z_{32}^* - \sqrt{2}Z_{22}^*$ and $\sqrt{2}Z_{23}^* - Z_{33}^*$ are calculated and their signs and magnitudes are expressed as vectors in Fig. 3c, d, respectively. As a result, the vectors of the pairs of planes {L1, L2} and {L3, L4} have opposite signs to each other which allows the generation of LO phonons with amplitudes proportional to the differences in the values ($Z_{32}^* - \sqrt{2}Z_{22}^*$ and $\sqrt{2}Z_{23}^* - Z_{33}^*$). However, the pair of adjacent planes {L2, L3} have the same sign, implying that the LO phonons of the pairs of planes {L1, L2} and {L3, L4} have opposite phase to each other. Therefore, the $Z_{32}^* - \sqrt{2}Z_{22}^*$ term, where the two LO phonons have nearly the same amplitude, will not produce a macroscopic reflectance change due to the cancellation of the two contributions. On the other hand, the terms with the values $\sqrt{2}Z_{23}^* - Z_{33}^*$ are expected to remain, since the amplitudes of the LO phonons in the plane pairs {L1, L2} are much larger than those in {L3, L4} (Fig. 3d). Moreover, since the value $\chi_{31}^{(2)} - \chi_{33}^{(2)}$ (generally $\chi_{31}^{(2)} < \chi_{33}^{(2)}$) is not zero, the second term $(\sqrt{2}Z_{23}^* - Z_{33}^*)(\chi_{31}^{(2)} - \chi_{33}^{(2)})$ in Eq. (3) can dominate. Thus, the Born effective charge tensor $Z^*$ in the layer containing the vacancies is concluded to be the origin of the large driving force $\mathbf{F}_{NV}$.

Note that locally enhanced LO phonon motion occurring only around the NV centers cannot explain the >10-fold enhancement of both the electronic and phononic amplitudes. Since the density of the NV centers is only ≈$10^{16}$ cm$^{-3}$ for the dose level of $1.0 \times 10^{12}$ N$^+$ cm$^{-2}$ and is ≪ $10^{-6}$ compared with the atomic density of the adjacent carbon atoms,

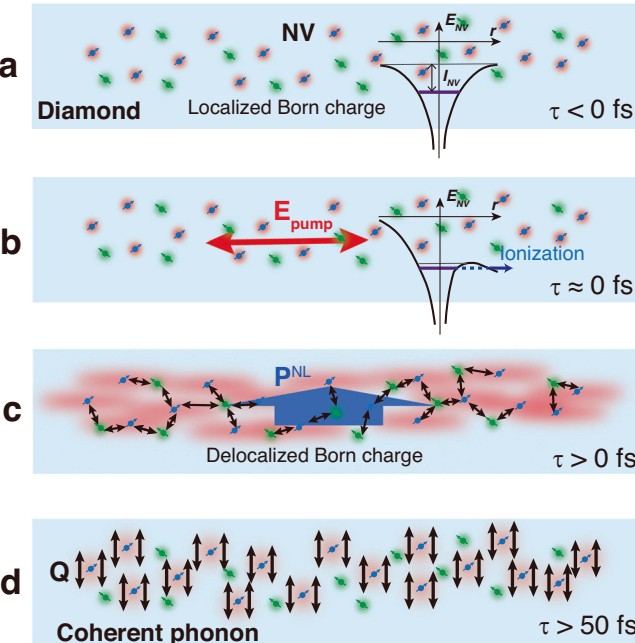

**Fig. 4 | Schematic presentation of the cooperative polaronic picture in NV diamond. a** NV diamond before excitation. The NV center is a mixture of NV⁻ (red charge distribution) and NV⁰ (green charge distribution) states. The inset represents a local potential around the NV center, where $r$ is distance from the NV center and $I_{NV}$ is the ionization energy. **b** Upon the photoexcitation the NV centers are photoionized by the pump electric field $\mathbf{E}_{pump}$, resulting in ionization. **c** Just after the photoexcitation, Born effective charges are strongly delocalized and spread over the distance of the NV centers, forming a nonlinear polarization field $\mathbf{P}^{NL}$, whose average generates the amplified driving force $\mathbf{F}_{NV}$. The red spheres indicate the long-range dipolar Fröhlich polaron. **d** The coherent LO phonons are driving by $\mathbf{F}_{NV}$.

the local LO phonon amplitude should be $\Delta a/a \sim 10^2$ based on the expected phonon amplitude in NI diamond of $\Delta a/a \sim 10^{-4}$, where $\Delta a$ is the phonon amplitude in real space[36]. This huge amplitude ($\Delta a/a \sim 10^2$) is, however, not expected to occur as it would exceed the lattice constant and exceed the Lindemann criterion[37]. The cosine-like driving force implies the presence of step-like nonlinear polarization and near-resonant conditions. Since the electric field of the pump light exceeds $1.4 \times 10^9$ V m$^{-1}$ or $1.4 \times 10^7$ V cm$^{-1}$, field-induced ionization of NV⁻ center is expected to occur. Thus, the Born effective charge around the NV⁻ centers will be delocalized under nonequilibrium conditions through field-induced ionization. In fact, at our experimental intensities, the pump light field is expected to inject electrons into the conduction bands by multiphoton ionization as suggested by the Keldysh parameter[38], $\gamma = \frac{\omega}{eE}\left(m_e^* I_{NV}\right)^{\frac{1}{2}} \approx 3$, as calculated for NV diamond, where $I_{NV}$ is the ionization energy of NV centers. However, the linear dependence of the phonon amplitude (Supplementary Fig. 3) suggests that field-induced tunneling ionization or Franz-Keldysh effect is more plausible in our experiment[39], the latter of which changes the optical absorption edge when a strong electric field is applied. The density of the P1 center is extremely low because our diamond sample was high-purity electronic grade in which only [N] <5 ppb was included. After the introduction of the NV centers, however, the density of the P1 center is expected to increase as the dose [N⁺] increases[40]. The high-density P1 centers are thought to act as scattering centers for carriers and phonons, and thus reducing the coherent phonon amplitude ($A$) and the photo-induced current effect (cosine-like phase) as demonstrated in Fig. 2. Note that the P1 center cannot break the inversion symmetry of diamond, and thus cannot contribute to a cooperative polaronic

effect, although it may partly contribute to photoexcitation via Urbach tails.

## Proposed cooperative polaronic effects

We propose that cooperative effects between NV centers play the main role in producing macroscopic second-order nonlinear polarization $\mathbf{P}^{NL}$ as schematically visualized in Fig. 4, similar to the observation of enhancement of superradiation from nanodiamonds[41], but fundamentally different than that occurring under equilibrium conditions. In addition, the polaron is produced by Fröhlich electron-phonon coupling via a LO phonon, which is enhanced by the Born effective charge. The polaronic quasiparticle appears at the lower dose level of $1.0 \times 10^{12}$ N⁺ cm$^{-2}$, indicating the presence of competing attractive and dissociative effects around the NV center, i.e., the higher the NV density, the larger the attractive force, while the larger the defect density, the stronger the electron-defect and phonon-defect scattering become.

In our proposed model for cooperative effects, the two defects NV⁻ and NV⁰ play the main role in the charge transfer between them. Namely, cooperative effects will be proportional to the product of their densities, i.e., [NV⁻][NV⁰]. To further investigate the cooperative effects, we measured the dose dependence of the photoluminescence (Supplementary Fig. 4), which showed increases in both the ZPL at 638 nm [NV⁻] and the background at 600 nm [NV⁰] as the dose increased. In addition, [NV⁻] and [NV⁰] are saturated at high doses (shown in the right panel of Supplementary Fig. 4). These facts suggest that [NV⁻][NV⁰] may not show a nonlinear dependence. Therefore, instead of the product [NV⁻][NV⁰], we present a simple model of the enhancement $A \propto$ [N⁺] and defect scattering $A \propto 1/$[N⁺] as shown in Fig. 2. As to why the largest EO and phonon signals were obtained at

$1 \times 10^{12} N^+ cm^{-2}$, we speculate that it may be a trade-off point between increased Born effective charge and scattering of carriers (ionized) due to defects. It is interesting to note that optically detected magnetic resonance (ODMR) measurements show the contrast of the $NV^-$ resonant dip was maximized at $1 \times 10^{12} N^+ cm^{-2}$, indicating the density of $NV^-$ was enhanced (see Supplementary Fig. 5). Thus, both ultrafast EO and ODMR measurements suggest that $NV^-$ defects play the main role in the enhancement of the coherent phonon amplitude. In semiconductors, thermal phonons are generated either by the emission of phonons via photoexcited carriers (intraband relaxation) or anharmonic phonon-phonon scattering (optical phonon relaxation into acoustic phonons)[42]. The time scales for those thermal phonons are picoseconds[42], and therefore will not impact transient cooperative polaronic effects occurring within 100 fs.

The polaron can propagate through the NV layer (40 nm deep and 28 nm wide[21]) by charge transfer between the $NV^-$ and $NV^0$ centers, a process that can occur over distances exceeding several nanometers[43]. In fact, with a pulse width of about 10 fs, the distance electrons can travel during the pulse can be estimated to be $450 \, cm^2 \, V^{-1} s^{-1} \times 10^7 \, V \, cm^{-1} \times 10 \times 10^{-15} s = 450 \, nm$, assuming a mobility of $450 \, cm^2 \, V^{-1} s^{-1}$ (ref. 44). This is well beyond the spacing between $NV^-$ and $NV^0$ centers ($\backsim 30$ nm) and results in instantaneous electron transfer. More importantly, under irradiation by intense femtosecond laser pulses, electric-field ionization enables much longer charge transfer lengths, which would exceed the average spacing of $NV^-$ and $NV^0$ centers, although further investigations are required to fully understand the process. This polaronic picture will provide a paradigm shift for the physics of color centers in diamonds for applications in quantum network sciences.

In conclusion, we present the EO response from NI and NV diamonds in the terahertz frequency region excited by sub-10 fs laser pulses. Both electronic and phononic responses exhibit dramatic intensity increases, and in particular a 13-fold magnification for the light dose level of $1.0 \times 10^{12} N^+ cm^{-2}$. We suggest that the physical mechanism is the generation of an additional driving force by polaronic cooperative second-order nonlinear polarization $\mathbf{P}^{NL}$ via long-range dipolar Fröhlich interaction around the NV center by non-zero Born effective charge, occurring under the influence of the strong electric fields induced by the laser pulse. These results pave the way for a new strategy of quantum sensing technologies based on 40 THz longitudinal lattice strain fields. Moreover, by utilizing the amplification of phonons at the nanoscale by the NV centers through controlled doping positions, a paradigm shift for designing phononic nanodevices may be realized, a path which cannot be realized by conventional phonon engineering via impurity doping.

## Methods

### Sample preparation
The samples were comprised of Element Six [001]-oriented electronic grade (EG) diamond single crystal, fabricated by chemical vapor deposition with impurity (nitrogen: [N], boron: [B]) levels of [N] <5 ppb, [B] <1 ppb, and the typical NV center concentration being less than 0.03 ppb. The sample size was $2.0 \, mm \times 2.0 \, mm \times 0.5 \, mm$ (thickness). NV diamonds with NV centers were prepared by implanting 30 keV nitrogen ions ($^{14}N^+$) into the sample followed by annealing at 900–1000 °C for 1 h in an Argon atmosphere. The $^{14}N^+$ ion dose was fixed at the following four levels: $2.0 \times 10^{11}$, $1.0 \times 10^{12}$, $5.0 \times 10^{12}$, and $2.0 \times 10^{13} N^+ cm^{-2}$. The NV centers are produced at a depth of about $\backsim 40$ nm with a production efficiency of about 10% (ref. 45).

### Ultrafast spectroscopy
The electronic response and the coherent phonons of the diamonds were measured by the electro-optic (EO) sampling method based on a reflective pump-probe scheme[19,20]. The light source is an ultrashort

pulse femtosecond oscillator (Element 2, Spectra-Physics), which generates ≤10 fs pulses with a center wavelength of $\backsim 800$ nm (1.55 eV) at a repetition rate of 75 MHz. The signal $\Delta R_{EO}/R_0$ for EO sampling is the anisotropic reflectance change, and probe light in the [100] direction is used to detect the maximum amplitude coherent phonons of the [001] diamond single crystal. The pump polarization is the $[\bar{1}10]$ direction. The time delay between the pump and probe light is modulated at a frequency of 4.5 Hz and an amplitude of 15 ps by an oscillating retroreflector (shaker) placed in the pump light path. The light is focused onto the sample by a 90-degree off-axis parabolic mirror with a focal length of 50.8 mm. Assuming the incident beam diameter to be 4 mm, pump light with the average power of 440 mW and probe light with 1 mW correspond to the fluences of $\approx 1.3 \, mJ \, cm^{-2}$ and $3 \, \mu J \, cm^{-2}$, respectively. The random noise of the signal on each scan is reduced by integrating the signal with a digital oscilloscope, and the number of accumulations is 5000.

### DFT simulations
The Vienna ab initio Simulation Package (VASP 6) code was used for all calculations[46]. Calculations were carried out using the GGA exchange functional (PBE for solids) and a $3 \times 3 \times 3$ Monkhorst-Pack k-point grid[47] in conjunction with projected augmented wave pseudopotentials[48] with a plane wave cutoff of 680 eV. Spin orbital coupling effects were included. The $NV^-$ center was constructed in a 64-atom supercell with the supercell dimensions fixed to that of the corresponding relaxed diamond structure. The energy convergence criterion was $10^{-8}$ eV. The Born effective charges within the supercell were determined using density functional perturbation theory within VASP.

## Data availability
All data that support the findings in this paper are available within the article and Supplementary Information. Any additional information can be available from the corresponding authors upon request. (Received: July 19, 2024.) Source data are provided with this paper.

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

## Acknowledgements

This research was supported by Grants-in-Aid for Scientific Research from the Japan Society for the Promotion of Science (JSPS) (Grant Nos. 22H01151 (M.H.), 22J11423 (T.I.), 22KJ0409 (T.I.), 23K22422 (M.H.), and 24K01286 (T.A.)), and by CREST, Japan Science and Technology Agency (Grant No. JPMJCR1875) (M.H.). T.I. acknowledges the support from a Grant-in-Aid for Japan Society for the Promotion of Science (JSPS) Fellows.

## Author contributions

T.I. and M.H. planned and organized this project. D.P. and T.A. fabricated the sample. T.I., J.G., and M.H. performed experiments and analysed the data. P.F. performed the DFT calculations. T.I., J.G., P.F., and M.H. discussed the results. T.I., P.F., and M.H. co-wrote the manuscript.

## Competing interests

The authors declare no competing interests.
