## [Peer Review File · Nature Communications]

REVIEWER COMMENTS

Reviewer #1 (Remarks to the Author):

This manuscript reports the observation of polarons involving the nitrogen-vacancy center of diamond, made using a combination of ultrafast optical pump-probe spectroscopy and electronic structure calculations. While the experimental data is of high quality, it is not clear to me that the experimental data supports the authors' claim that a long-range polaronic interaction around the NV center drives coherent phonon generation. Hence, I am unable to recommend publication of the present manuscript in Nature Communications. My specific comments are as follows.

(1) My main confusion lies in how the authors assign the mechanism of coherent phonon generation in NV diamond. The lowest-energy transition in NV diamond is at 638 nm. However, the spectral range of their laser pulse spans 660 – 940 nm, insufficient to excite the NV diamond. In this case, how were polarons produced when the excitation is nonresonant?

(2) The above leads to the next question of how nonresonant excitation could have given rise to a cosine-like driving force. Furthermore, the simulations do not address how an initial oscillation phase of $\pi/2$ might have been obtained.

(3) The authors obtain the dependence of the amplitude, frequency, and phase on the doping density. However, the origins of these density-dependent behavior are not explained. It is also puzzling how the authors could have discerned a frequency shift of 0.02 THz (Fig. 2d), when the range of time delay extends to only ~ 9 ps, which gives a frequency resolution of ~ 0.1 THz. Related to this point, it would be better if error bars could be added to the data points in Figs. 2b – 2e.

(4) The authors attribute the sample response at ~ 0 fs to phonons. How did they exclude the possibility of coherent artifacts giving rise to the observed EO response? Along these lines, it would be useful if the authors could show a long-scan autocorrelation measurement of the laser pulses, say, from -100 fs to +100 fs.

(5) The authors claim that the polaron can propagate by charge transfer between NV- and NV0 centers, a process that has been shown by theoretical studies to occur over length scales of a few nanometers. However, the density of NV centers in the sample under investigation is only $\sim 10^{16}$ cm⁻³, implying that adjacent NV centers are tens of

nanometers apart. As such, it is unclear if the said charge transfer process can be operative.

(6) The authors propose that cooperative effects between NV centers are expected to give rise to a macroscopic second-order nonlinear polarization. However, it is not clear if this claim is supported by their experimental data. For cooperative effects to be observed, some nonlinear dependence on the defect concentration should be present. However, such a dependence is not shown.

Reviewer #2 (Remarks to the Author):

The authors present EO response from NI and NV diamonds in the terahertz frequency region excited by sub-10 fs laser pulses. The authors observed 13-fold enhancement of phononic responses on the nanoscale by NV center, which could lead to a transformative path forward for phonon-based quantum engineering. The manuscript is well. My concerns are listed below.

Major concern:

1. The authors claim that the enhancement they observed is due to a combination of cooperative polaronic effect and scattering by defects. However, the manuscript, as written, does not provide sufficient evidence.

a. At high defect density, the coherent modulation decays due to a larger defect density. What happens to much lower densities like $<1 \times 10^{12}$. If it is a competing attractive and dissociative effect, one could anticipate that coherent phonon modulation amplitude would decay again at a much lower density of NV.

2. The concept of polaron is a free carrier accompanied by phonon clouds. However, NV center is a Deep band gap state located over 2 eV below the conduction band. So, in normal circumstances, they are bound within atomic sites (PHYSICAL REVIEW LETTERS 120, 136401 (2018)). However, the authors claim the polaron can propagate over 40 nm by charge transfer from NV⁻ to NV⁰. However, the manuscript failed to provide sufficient

detailed explanation or experimental and/or theoretical evidence of the mechanism that charge carriers located over 2eV below the conduction band and not significantly excited by the 800 nm pump pulse able to move 40 nm. Since the NV density is $\sim 1 \times 10^{16} \text{ cm}^{-3}$, which is very low, with an average NV-to-NV distance of 20 nm, tunneling or hopping probability is extremely low. The cited reference (Ref 33, fig4) estimated the rate of hopping as almost negligible with an average NV-to-NV distance of 5 nm.

3. What is the NV0 and NV- density in the samples? Since NV0 and NV- are very different defects, NV- contains one extra electron compared to NV0. The electronic wave function is also very different for these two types of defects. How does NV- vs NV0 contribute to a cooperative polaronic effect?

4. How do the other defect centers, such as the P1 center, impact the cooperative polaronic effect?

Minor concern.

1. "Photoluminescence measurements indicate the electronic state of the NV diamond is a mixture of negatively charged states" Needs reformatting. ZPL for NV- is 638 and peaks at ~ 680 nm. ZPL for NV0 at 575 and peak at ~ 660 nm.

2. The incident pump energy is 800 nm (1.55 eV) to excite LO phonons (0.165eV). Huge energy is given to the crystal, which may produce large quantities of thermal phonons. Can the author comment on how this excess energy impacts the cooperative polaronic effect?

3. The incident pump power is 440 mw over 2 mm, which is large. Can the author comment on why they chose this pump power? How cooperative polaronic effect depends on laser pump power.

4. At this high pump power, there is a possibility for multiphoton excitation of NV centers (Phys. Rev. B 97, 134112 (2018)). Can the author make comments on the estimated rate for multiphoton ionization at this pump power and how that impacts the cooperative polaronic effect?

Response to Reviewer #1

We would like to express our gratitude to the reviewers for reading our manuscript and providing fruitful comments. We have conducted additional experiments after receiving the initial decision of the editors and have revised our manuscript following the reviewers' suggestions.

[Referee] This manuscript reports the observation of polarons involving the nitrogen-vacancy center of diamond, made using a combination of ultrafast optical pump-probe spectroscopy and electronic structure calculations. While the experimental data is of high quality, it is not clear to me that the experimental data supports the authors' claim that a long-range polaronic interaction around the NV center drives coherent phonon generation. Hence, I am unable to recommend publication of the present manuscript in Nature Communications. My specific comments are as follows.

(1) My main confusion lies in how the authors assign the mechanism of coherent phonon generation in NV diamond. The lowest-energy transition in NV diamond is at 638 nm. However, the spectral range of their laser pulse spans 660 – 940 nm, insufficient to excite the NV diamond. In this case, how were polarons produced when the excitation is nonresonant?

#Our reply#

Both reviewers expressed concerns that our laser energy (wavelength) may be insufficient to generate free carriers from the excitation of NV states. Our reply is as follows. First, there is possible excitation of NV states via the Urbach tail associated with the presence of other defect states, such as P1 centers, which will induce a Urbach tail for the NV level with > 1 eV broadening as shown in the density of states calculated by density functional theory (DFT) (Fig. R1 below), enabling the absorption of laser light even when the spectra extends up to 660 nm. The density of the P1 center is extremely low because our diamond sample was high purity electronic grade in which only $[N] < 5$ ppb was included. After the introduction of the NV centers, however, the density of the P1 center is expected to increase as the dose $[N^+]$ increases (Park, H., Lee, J., Han, S. *et al.* npj Quantum Inf. **8**, 95 (2022)). The high density P1 centers will act as scattering center for carriers and phonons, and thus reduce the coherent phonon amplitude (A) and photo-induced current effect (cosine-like phase). Note that the P1 center cannot break the inversion symmetry of diamond, and thus cannot contribute to the cooperative polaronic effect, although it may partly contribute to the photoexcitation via Urbach tails.

Fig. R1. DFT calculations of density of states (DOS) for the NV diamond using VASP code with 64 atoms supercell. The left panel present wider energy range, while the right panel focuses in the vicinity of the band gap. NV-1 represent the DOS for NV^- center and P1 represents the DOS for the P1 center. The right panel shows an enlargement around the band gap.

In the present case, more importantly, the electric field of the pump light (440 mW) is 1.4×10^9 V/m or 1.4×10^7 V/cm, which is close or comparable to the dielectric breakdown threshold ($1-2 \times 10^7$ V/cm), and is sufficient to cause transient field-induced ionization of NV^- centers, please refer e.g., P. Rodin *et al.*, J. Appl. Phys. Vol.98, 094506 (2005). Thus, the Born effective charge around the NV^- centers will be delocalized under the nonequilibrium conditions brought by field-induced ionization. In fact, at our experimental intensities, the light field is expected to inject electrons into the conduction bands by multiphoton ionization as suggested by the Keldysh parameter (please refer e.g., Schaffer and Mazur, Meas. Sci. Technol. **12** (2001) 1784–1794; M. Kozák *et al.*, Phys. Rev. B. **99** (2019) 104305), value of $\gamma \approx 3$, as calculated for our experimental conditions. However, the linear dependence of the phonon amplitude in Fig.R2 suggests that field-induced tunneling ionization or the Franz-Keldysh effect are the more plausible generators of free carriers in our experiment (Lucchini *et al.*, Science **353**, 916 (2016)), the latter of which leads to changes in the optical absorption edge when a strong electric field is applied. In the revision, we have accordingly added discussions regarding these effects.

Fig. R2. The amplitude of coherent LO phonons as a function of the pump power for NI and NV ($1 \times 10^{12} N^+ cm^{-2}$) diamonds. The solid lines were fit using a linear function. The maximum power for these additional measurements was 430 mW, slightly below that of the main text due to the laser condition.

[Referee]

(2) The above leads to the next question of how nonresonant excitation could have given rise to a cosine-like driving force. Furthermore, the simulations do not address how an initial oscillation phase of $\pi/2$ might have been obtained.

#Our reply#

As mentioned above, the application of a strong electric field causes photo-induced current and is equivalent to normal electronic excitation by visible light and coherent phonons are generated. Since electronic excitation is taking place, the initial phase is of cosine form. Note that a cosine-like driving force under near-resonant excitation conditions was also observed in a *n*-doped Si crystal; here the E_0 resonance of 3.4 eV (= 365 nm) was observed, for a laser energy of 2.91 – 3.26 eV (380 – 425 nm) [M. Hase *et al.*, Nature **426**, p.51 (2003); New J. Phys. **14**, p.055018 (2013)], where the possible photoexcitation of carriers using light near-resonance was thought to occur via semiconducting-like Urbach tails; this is not the case here. Lastly, a DFT simulation was performed for the equilibrium state before photoexcitation. A simulation for the non-equilibrium state will require time-dependent DFT, which is a difficult task for us at present. We have explained in the revised Fig. 4 how the coherent phonons are excited as well presence of the cosine-like phase.

[Referee]

(3) The authors obtain the dependence of the amplitude, frequency, and phase on the doping density. However, the origins of these density-dependent behavior are not explained. It is also puzzling how the authors could have discerned a frequency shift of 0.02 THz (Fig. 2d), when the range of time delay extends to only ~ 9 ps, which gives a frequency resolution of ~ 0.1 THz. Related to this point, it would be better if error bars could be added to the data points in Figs. 2b – 2e.

#Our reply#

Thank you for the suggestion on the accuracy of the fit. We agree with the reviewer's statement that the frequency resolution is ~ 0.1 THz. In the revision, we have added error bars in Figs. 2b-2e accordingly. In addition, we have added a discussion for the density-dependent behavior of the amplitude, frequency, and phase in the revised manuscript.

[Referee]

(4) The authors attribute the sample response at ~ 0 fs to phonons. How did they exclude the possibility of coherent artifacts giving rise to the observed EO response? Along these lines, it would be useful if the authors could show a long-scan autocorrelation measurement of the laser pulses, say, from -100 fs to +100 fs.

#Our reply#

Coherent artifacts are frequently observed at $\tau \approx 0$ fs in pump-probe experiments, e.g., Opt. Exp. 17, 11321 (2009); Phys. Rev. Lett. 102, 117201 (2009). We observed coherent artifacts on our EO signal as can be seen in Fig. R3 below, showing $\tau \approx 0$ fs is slightly before the peak and continues to spike up to ~ 100 fs.

The coherent phonon signal was measured by maximizing the phonon signal while the initial EO response was maxed out. Then, coherent phonon signal was extracted from the EO response using bandpass filtering, since the electronic response and phonon response are well separated as shown in Fig. 1(b). The time-domain fit shown in Fig. 2(a) was performed after ≈ 100 fs to avoid effects from coherent artifacts. Thus, the problematic coherent artifacts were excluded in the fit of coherent phonon signal in Fig. 2(a). In the revision, we have mentioned the effects of coherent artifacts in the determination of the real $\tau \approx 0$ fs position, and explained the reason why we fit the coherent phonon signal from $\tau \approx 100$ fs.

Fig. R3. Time-resolved EO response around $\tau = 0$, demonstrating the appearance of coherent artifacts at $\tau = 0$. The top fringe-like coherent artifacts were extracted from the EO response for the NI diamond.

[Referee]

(5) The authors claim that the polaron can propagate by charge transfer between NV- and NV0 centers, a process that has been shown by theoretical studies to occur over length scales of a few nanometers. However, the density of NV centers in the sample under investigation is only $\sim 10^{16} \text{ cm}^{-3}$, implying that adjacent NV centers are tens of nanometers apart. As such, it is unclear if the said charge transfer process can be operative.

#Our reply#

The electric field of the pump light was $1.4 \times 10^9 \text{ V/m}$ or $1.4 \times 10^7 \text{ V/cm}$, which is sufficient to induce field-induced ionization of the NV^- centers. Thus, the Born effective charge around the NV^- centers will be delocalized under nonequilibrium conditions through the field-induced ionization. On the femtosecond time scale, electrons (carriers) are largely delocalized. In fact, the pulse width is about 10 fs, and the distance electrons can travel within this time is $450 \text{ cm}^2/\text{Vs} \times 10^7 \text{ V/cm} \times 10 \times 10^{-15} \text{ s} = 450 \text{ nm}$, assuming a mobility of $450 \text{ cm}^2/\text{Vs}$ at room temperature (E. Bourgeois *et al.*, *Nature Commun.* **6**, 8577 (2015)). This is well beyond the average NV–NV ($\sim 30 \text{ nm}$) spacing and results in instantaneous electron transfer. In the revised manuscript, we have added these points accordingly.

[Referee]

(6) The authors propose that cooperative effects between NV centers are expected to give rise to a macroscopic second-order nonlinear polarization. However, it is not clear if this claim is supported by their experimental data. For cooperative effects to be observed, some nonlinear dependence on the defect concentration should be present. However, such a dependence is not shown.

#Our reply#

To investigate a possible nonlinear dependence on the defect concentration, we have added data for the lower dose of $2 \times 10^{11} \text{ N}^+ \text{ cm}^{-2}$ as shown in the revised Fig. 2. The results show an amplitude reduction by a factor of 5 for the $2 \times 10^{11} \text{ N}^+ \text{ cm}^{-2}$ dose sample. In our proposed explanation of cooperative effects, the two particles from the NV- and NV0 centers play the main role in the charge transfer between them. Namely, cooperative effects will be proportional to the product of the density, i.e., $[\text{NV}^-][\text{NV0}]$. To further investigate the cooperative effects, we measured the dose dependence of photoluminescence as shown in Fig. R4 below, which shows the increase in both the ZPL at 638 nm [NV-] and background at 600 nm [NV0] as the dose increases. In addition, [NV-] and [NV0] saturated for high doses, as shown in the right panel. These facts suggest that $[\text{NV}^-][\text{NV0}]$ may not show a nonlinear dependence. In addition, [NV-] and [NV0] are in equilibrium with each other (meaning they can easily switch between each state: please refer, e.g., Y. Doi *et al.* *Phys. Rev. B* **93**, 081203(R) (2016)), implying it is difficult to determine the densities for [NV-] and [NV0] on ultrafast time scales. Instead of the product $[\text{NV}^-][\text{NV0}]$, we have presented a simple model for the enhancement $A \propto [\text{N}^+]$ and defect scattering $A \propto 1/[\text{N}^+]$ in the revised version of Fig. 2(b).

Fig. R4. (Left) Photoluminescence spectra obtained for diamond samples using 532 nm cw laser using a fiber coupled visible spectrometer. (Right) The PL intensity as the function of the N^+ dose for the zero phonon line (ZPL) at 638 nm of the NV^- center and the background component at 600 nm reflecting the NV^0 center.

Response to Reviewer #2:

We are grateful to the reviewers for reading our manuscript and their fruitful comments. We have conducted additional experiments after receiving the initial editor decision and now have revised the manuscript following the reviewer's suggestions.

[Referee]

The authors present EO response from NI and NV diamonds in the terahertz frequency region excited by sub-10 fs laser pulses. The authors observed 13-fold enhancement of phononic responses on the nanoscale by NV center, which could lead to a transformative path forward for phonon-based quantum engineering. The manuscript is well. My concerns are listed below.

Major concern:

1. The authors claim that the enhancement they observed is due to a combination of cooperative polaronic effect and scattering by defects. However, the manuscript, as written, does not provide sufficient evidence.

a. At high defect density, the coherent modulation decays due to a larger defect density. What happens to much lower densities like $<1 \times 10^{12}$. If it is a competing attractive and dissociative effect, one could anticipate that coherent phonon modulation amplitude would decay again at a much lower density of NV.

#Our reply#

Thank you for the suggestion of additional experiments. We have fabricated an additional NV diamond sample with a dose of $2 \times 10^{11} \text{ N}^+ \text{ cm}^{-2}$ to determine if the coherent phonon modulation amplitude decays similarly at a much lower density. The results are shown below (Fig. R5). As can be seen in Fig. R5, the coherent phonon amplitude for the lower density of $2 \times 10^{11} \text{ N}^+ \text{ cm}^{-2}$ sample is smaller than that of $1 \times 10^{12} \text{ N}^+ \text{ cm}^{-2}$ sample, suggesting that the $1 \times 10^{12} \text{ N}^+ \text{ cm}^{-2}$ sample exhibits maximum amplification. Thus, we have revised Fig. 2 in the revised manuscript.

Fig. R5. (revised Fig. 2) Time-domain $\Delta R_{EO}/R_0$ signals for NI and NV diamonds obtained at the pump fluence of 1.3 mJ cm^{-2} . We have added new data for lower dose of $2 \times 10^{11} \text{ N}^+ \text{ cm}^{-2}$.

[Referee]

2. The concept of polaron is a free carrier accompanied by phonon clouds. However, NV center is a Deep band gap state located over 2 eV below the conduction band. So, in normal circumstances, they are bound within atomic sites (PHYSICAL REVIEW LETTERS 120, 136401 (2018)). However, the authors claim the polaron can propagate over 40 nm by charge transfer from NV⁻ to NV⁰. However, the manuscript failed to provide sufficient detailed explanation or experimental and/or theoretical evidence of the mechanism that charge carriers located over 2eV below the conduction band and not significantly excited by the 800 nm pump pulse able to move 40 nm. Since the NV density is $\sim 1 \times 10^{16} \text{ cm}^{-3}$, which is very low, with an average NV-to-NV distance of 20 nm, tunneling or hopping probability is extremely low. The cited reference (Ref 33, fig4) estimated the rate of hopping as almost negligible with an average NV-to-NV distance of 5 nm.

#Our reply#

As reviewer#1 also expressed concerns that our laser energy (wavelength) insufficient to excite to generate free carriers from NV states, we address this issue below. First, there is the possible excitation of NV states via Urbach tails associated with the presence of other defect states, such as P1 centers, which will induce an Urbach tail for the NV level with ~ 1 eV broadening as shown in the density of states calculated by DFT (Fig. R1), enabling the absorption of laser light even when the spectra extends up to 660 nm. The density of the P1 center is extremely low because our diamond sample was high purity electronic grade in which only $[N] < 5$ ppb was included. After the introduction of the NV centers, however, the density of the P1 center would increase as the dose $[N^+]$ increases (Park, H., Lee, J., Han, S. *et al.* npj Quantum Inf. **8**, 95 (2022)). The high density of P1 centers will become scattering centers for carriers and phonons, and thus reduce the coherent phonon amplitude (A) and photo-induced current effect (cosine-like phase). Note that the P1 center cannot break the inversion symmetry of diamond, and thus cannot contribute to the cooperative polaronic effect, although it may partly contribute to photoexcitation via Urbach tails.

Fig. R1. DFT calculations of density of states (DOS) for the NV diamond using VASP code with 64 atoms supercell. The left panel present wider energy range, while the right panel focuses in the vicinity of the band gap. NV-1 represent the DOS for NV⁻ center and P1 represents the DOS for the P1 center. The right panel shows an enlargement around the band gap.

In the present case, more importantly, the electric field of the pump light (440 mW) is 1.4×10^9 V/m or 1.4×10^7 V/cm, which is close to or exceeds the dielectric breakdown threshold ($1-2 \times 10^7$ V/cm), and is sufficient to cause field-induced ionization of NV⁻ centers. Thus, the Born effective charge around the NV⁻ centers is expected to be delocalized under nonequilibrium conditions through field-induced ionization. In fact, at our experimental intensities, the IR field is also expected to inject electrons into the conduction bands by multiphoton ionization as suggested by

the Keldysh parameter (please refer e.g., Schaffer and Mazur, *Meas. Sci. Technol.* **12** (2001) 1784–1794; M. Kozák *et al.*, *Phys. Rev. B.* **99** (2019) 104305), $\gamma \approx 3$, which for our case. However, the linear dependence of phonon amplitude in Fig.R2 suggests that field-induced tunneling ionization or the Franz-Keldysh effect is a more plausible explanation for our experiment (Lucchini *et al.*, *Science* **353**, 916 (2016)), the latter of which changes the optical absorption edge when a strong electric field is applied.

As mentioned above, for femtosecond time scales, electrons (carriers) are largely delocalized. In fact, the pulse width is about 10 fs, and the distance electrons can travel within this time is $450 \text{ cm}^2/\text{Vs} \times 10^7 \text{ V/cm} \times 10 \times 10^{-15} \text{ s} = 450 \text{ nm}$, assuming a mobility of $450 \text{ cm}^2/\text{Vs}$ at room temperature (E. Bourgeois *et al.*, *Nature Commun.* **6**, 8577 (2015)). This is well beyond the NV–NV (~30 nm) spacing and results in an instantaneous electron transfer. In the revised manuscript, we have added these points accordingly.

[Referee]

3. What is the NV0 and NV- density in the samples? Since NV0 and NV- are very different defects, NV- contains one extra electron compared to NV0. The electronic wave function is also very different for these two types of defects. How does NV- vs NV0 contribute to a cooperative polaronic effect?

#Our reply#

Thank you for this important question. In our proposal for cooperative effects, two particles from NV- and NV0 centers will play the main role in the charge transfer between the centers. Namely, the cooperative effects will be proportional to the product of the densities, i.e., $[\text{NV-}][\text{NV0}]$. We have also investigated the dose dependence of the photoluminescence as shown Fig. R4 below, which show an increase in both the ZPL at 638 nm [NV-] and the background at 600 nm [NV0] as the dose increases. In addition, [NV-] and [NV0] saturate for high doses, as shown in the right panel. These facts suggest that $[\text{NV-}][\text{NV0}]$ will not show the nonlinear dependence. In addition, [NV-] and [NV0] are in equilibrium with each other (meaning they can easily switch between states: please refer, e.g., Y. Doi *et al.* *Phys. Rev. B* **93**, 081203(R) (2016)), implying it is difficult to determine the densities for [NV-] and [NV0] on ultrafast time scales. Instead of the product $[\text{NV-}][\text{NV0}]$, we have introduced a simple model for the enhancement $A \propto [\text{N+}]$ and defect scattering $A \propto 1/[\text{N+}]$ in the revised version of Fig. 2(b).

We have also investigated the density of the NV- centers using photoluminescence measurements, finding an increase in intensity as the N+ dose increases. Based on our model, Eq. (1), NV- is the core defect that promotes cooperative polaronic effects, since only NV- has an extra electron and the charge distribution generates Born effective charges around the NV- center, resulting in the polaronic driving force of Eq. (1). DFT simulations indicate the Born effective charges are nearly negligibly small for the case of NV0 centers.

Fig. R4. (Left) Photoluminescence spectra obtained for the diamond samples using 532 nm cw laser using a fiber coupled visible spectrometer. (Right) PL intensity as the function of N^+ dose for zero phonon line (ZPL) at 638 nm of the NV- center and the background component at 600 nm reflecting the NV⁰ center.

Furthermore, we investigated why the dose of $1 \times 10^{12} N^+ cm^{-2}$ led to largest coherent phonon signal. As presented in Fig. R6 below, the ODMR measurements show the contrast of the NV- resonant dip was maximized at $1 \times 10^{12} N^+ cm^{-2}$, indicating the density of NV- was enhanced. Thus, both ultrafast EO and ODMR measurements support the premise that NV- plays the main role in the enhancement of the coherent phonon amplitude. In the revision, we have added these observations to the discussion accordingly. Thus, we can conclude that NV- centers play the major role.

Fig. R6. Optically detected magnetic resonance (ODMR) spectra obtained at room temperature for the NV diamond samples using 532 nm cw laser.

[Referee]

4. How do the other defect centers, such as the P1 center, impact the cooperative polaronic effect?

#Our reply#

Thank you for this important question. The P1 center consists of N atoms substituted at C atom sites. The density of the P1 center is extremely low because our diamond sample was high purity electronic grade in which only $[N] < 5$ ppb was included. After the introduction of the NV centers, however, the density of the P1 center is expected to increase as the dose $[N^+]$ increases (Park, H., Lee, J., Han, S. *et al.* npj Quantum Inf. **8**, 95 (2022)). The high density of P1 centers will become scattering centers for carriers and phonons, and thus reduce the coherent phonon amplitude (A) and photo-induced current effect (cosine-like phase). Note that the P1 center cannot break the inversion symmetry of diamond, and thus cannot contribute to the cooperative polaronic effect, although it

would partly contribute to the photoexcitation via Urbach tails. In the revised version, we have added a note regarding this point.

[Referee]

Minor concern.

1. “Photoluminescence measurements indicate the electronic state of the NV diamond is a mixture of negatively charged states” Needs reformatting. ZPL for NV⁻ is 638 and peaks at ~680 nm. ZPL for NV⁰ at 575 and peak at ~660 nm.

#Our reply#

We thank the review for pointing out the grammar issue. In the revision, we have revised this part accordingly.

[Referee]

2. The incident pump energy is 800 nm (1.55 eV) to excite LO phonons (0.165eV). Huge energy is given to the crystal, which may produce large quantities of thermal phonons. Can the author comment on how this excess energy impacts the cooperative polaronic effect?

#Our reply#

In semiconductors, thermal phonons are generated either by the emission of phonons via photoexcited carriers (intraband relaxation) or anharmonic phonon-phonon scattering (optical phonon relaxation into acoustic phonons)[Please refer e.g., S. K. Sundaram & E. Mazur, Nature Materials, Vol.1, 217, (2002). The time scales for those thermal phonons are picoseconds (> 1 ps), much slower than the EO response (< 100 fs), and therefore will not impact on the transient cooperative polaronic effect occurring within 100 fs. In the revised manuscript, we have added these points accordingly.

[Referee]

3. The incident pump power is 440 mw over 2 mm, which is large. Can the author comment on why they chose this pump power? How cooperative polaronic effect depends on laser pump power.

#Our reply#

The pump power of 440 mW was the largest available in our pump-probe set up and it corresponds to an electric field of 1.4×10^9 V/m or 1.4×10^7 V/cm, which is comparable to the dielectric breakdown threshold ($1-2 \times 10^7$ V/cm), and is sufficient to cause field-induced ionization of NV⁻ centers. We have observed a nearly linear dependence in the phonon response for the EO signal as a function of the pump power from 50 to 430 mW, as presented in Fig. R2 below. Although the maximum amplification is less than that reported in present manuscript, additional experiment on the pump power dependence suggests that the cooperative polaronic effect, that is the enhancement, is already present at the lowest power of 50 mW, although the amplification factor was ~ 4. Although more experimental and theoretical studies are required, the present data support the idea that the cooperative polaronic effect originates from field-induced ionization, i.e., the higher the pump power, the larger the photoinduced charge current which in turn drive coherent LO phonon generation. In the revised manuscript, we have added these points accordingly.

Fig. R2. The amplitude of coherent LO phonon as a function of the pump power for the NI and NV ($1 \times 10^{12} \text{ N}^+ \text{ cm}^{-2}$) diamonds. The solid lines are the fit using a linear function. The maximum power in the additional measurements was 430 mW, slightly below that of the main text due to the laser conditions.

[Referee]

4. At this high pump power, there is a possibility for multiphoton excitation of NV centers (Phys. Rev. B 97, 134112 (2018)). Can the author make comments on the estimated rate for multiphoton ionization at this pump power and how that impacts the cooperative polaronic effect?

#Our reply#

Thank you for this important question. At our experimental intensities, the IR field is also expected to inject electrons into the conduction bands by multiphoton ionization as suggested by the Keldysh parameter (please refer e.g., Schaffer and Mazur, Meas. Sci. Technol. **12** (2001) 1784–1794; M. Kozák *et al.*, Phys. Rev. B. **99** (2019) 104305), $\gamma \approx 3$ for our case. However, the linear dependence of phonon amplitude in Fig.R2 suggests that field-induced tunneling ionization or Franz-Keldysh effect is more plausible in our experiment (Lucchini *et al.*, Science **353**, 916 (2016)), the latter of which changes the optical absorption edge when a strong electric field is applied. In the revised manuscript, we have added these points accordingly.

Summary of the changes made:

1. Page 3 second paragraph and new Fig. S1: We have emphasized the possible excitation of carriers through the field-induced ionization even under non-resonant conditions. (Response to the 1st comment from the Reviewer #1 and the 2nd major comments from the Reviewer #2.)
2. Page 3 second paragraph: We have fixed a grammatical issue. (Response to the 1st minor comment from the Reviewer #2.)
3. Page 4 first paragraph and new Fig. S2: We have added an explanation about coherent artifacts. (Response to the 4th comment from the Reviewer #1.)
4. Page 4~Page 5 first paragraph: We have added an explanation for the dose dependence of the obtained parameters in Fig. 2. (Response to the 3rd comment from the Reviewer #1.)
5. Page 5~6 and Eq. (2): We have added explanation for how non-resonant excitation gave rise to a cosine-like driving force. We have also made corrections to Eq. (2) to include the transient photocurrent contribution, which acts as the cosine-like driving force. (Response to the 2nd comment from the Reviewer #1.)
6. Page 8 first paragraph: We have added an explanation for the possible mechanism for cooperative polaronic effects based on transient delocalization of charge distribution. (Response to the 5th comment from the Reviewers #1, and the 2nd major comment from the Reviewer #2.)
7. Page 8 first paragraph: We have added a note regarding possible impacts of P1 center on the cooperative polaronic effect. (Response to the 4th major comment from the Reviewer #2.)
8. Page 9 second paragraph: We have added a discussion regarding the NV⁰ and NV⁻ densities in the samples, and the relevant contributions to cooperative polaronic effects. (Response to the 6th comment from the Reviewers #1, and 3rd major comment from the Reviewer #2.)
9. Page 9 second paragraph: We have added a discussion regarding the possibility of thermal phonon excess energy impacting the cooperative polaronic effects. (Response to the 2nd minor comment from the Reviewers #2.)
10. Page 10 second paragraph: We have added a discussion for the reason why charge transfer between NV⁻ and NV⁰ centres is possible. (Response to the comment from the 5th comment from Reviewer #1 and 2nd major comment from the Reviewer #2.)
11. The time axis of Fig. 1(a) has been corrected based on the presence of coherent artifacts. (Response to the 4th comment from the Reviewer #1.)
12. Fig. 2 was revised to add the data for lower dose of $2 \times 10^{11} \text{ N}^+ \text{ cm}^{-2}$, and to replace all the parameters after fitting with new time axis. (Response to the 1st major comment from the Reviewer #2.)
13. Fig. 3 was revised to fix typo.
14. Fig. 4 was revised to show time lapse of possible driving of coherent phonon by delocalized carrier excitation.

15. We have revised the format based on the guidelines including references.

REVIEWERS' COMMENTS

Reviewer #1 (Remarks to the Author):

The authors have thoroughly addressed the reviewers' comments, even going to the extent of performing additional experiments. I recommend publication of the manuscript in its current form.

Reviewer #2 (Remarks to the Author):

The authors provide very kind and complete answers to the questions. Some of them are still unclear to me. However, I think that it is usually impossible to understand/explain all experiment results in one manuscript. The revised manuscript has significant value for an article. So, I would recommend publication.